# Early-Onset Ovarian Cancer <30 Years: What Do We Know about Its Genetic Predisposition?

**DOI:** 10.3390/ijms242317020

**Published:** 2023-11-30

**Authors:** Klara Horackova, Marketa Janatova, Petra Kleiblova, Zdenek Kleibl, Jana Soukupova

**Affiliations:** 1Institute of Medical Biochemistry and Laboratory Diagnostics, First Faculty of Medicine, Charles University and General University Hospital in Prague, 128 00 Prague, Czech Republic; klara.horackova@lf1.cuni.cz (K.H.); mjana@lf1.cuni.cz (M.J.); pekleje@lf1.cuni.cz (P.K.); zdekleje@lf1.cuni.cz (Z.K.); 2Institute of Biology and Medical Genetics, First Faculty of Medicine, Charles University and General University Hospital in Prague, 128 00 Prague, Czech Republic; 3Institute of Pathological Physiology, First Faculty of Medicine, Charles University, 128 00 Prague, Czech Republic

**Keywords:** ovarian cancer, early-onset, genetic predisposition, germline pathogenic variant

## Abstract

Ovarian cancer (OC) is one of the leading causes of cancer-related deaths in women. Most patients are diagnosed with advanced epithelial OC in their late 60s, and early-onset adult OC diagnosed ≤30 years is rare, accounting for less than 5% of all OC cases. The most significant risk factor for OC development are germline pathogenic/likely pathogenic variants (GPVs) in OC predisposition genes (including *BRCA1*, *BRCA2*, *BRIP1*, *RAD51C*, *RAD51D*, Lynch syndrome genes, or *BRIP1*), which contribute to the development of over 20% of all OC cases. GPVs in *BRCA1*/*BRCA2* are the most prevalent. The presence of a GPV directs tailored cancer risk-reducing strategies for OC patients and their relatives. Identification of OC patients with GPVs can also have therapeutic consequences. Despite the general assumption that early cancer onset indicates higher involvement of hereditary cancer predisposition, the presence of GPVs in early-onset OC is rare (<10% of patients), and their heritability is uncertain. This review summarizes the current knowledge on the genetic predisposition to early-onset OC, with a special focus on epithelial OC, and suggests other alternative genetic factors (digenic, oligogenic, polygenic heritability, genetic mosaicism, imprinting, etc.) that may influence the development of early-onset OC in adult women lacking GPVs in known OC predisposition genes.

## 1. Introduction

Ovarian cancer (OC) accounts for 4.7% of cancer-related deaths in women worldwide [1]. Early detection of OC remains challenging due to the predominance of non-specific symptoms that occur mainly at advanced clinical stages [2]. The majority of patients are diagnosed with advanced disease with an unfavorable prognosis (5-year survival rate of approximately 50%), which is even worse in cases with metastatic disease (5-year survival rate of approximately 30%) [3]. Thus, it is important to identify women at increased OC risk early, especially when we know that the proportion of hereditary OC is high, reaching even over 20% [4,5,6]. Surprisingly, despite all assumptions, the abundance of germline pathogenic/likely pathogenic variant (GPV) carriers among early-onset OC patients (<30 years) falls below 10% [7,8,9], and the genetic predisposition to early-onset OC remains uncertain.

This review focuses on the characterization of early-onset adult OC patients (diagnosed between 18 and 30 years old) from the perspective of cancer predisposition, with a special focus on epithelial OC.

## 2. Characteristics of Early-Onset and Late-Onset OC

Ovarian cancer is a heterogeneous group, including tumors in the ovaries, fallopian tubes, and peritoneum. Approximately 90% of OC cases are of epithelial origin. The remaining 10% of OC are non-epithelial tumors consisting of germ cell, sex cord-stromal, and other rare non-epithelial ovarian tumors (including sarcoma or small cell carcinoma).

Epithelial OC can be stratified into five major histologic subtypes, including high-grade serous carcinoma (HGSC; accounting for up to 70% of all epithelial OC cases), endometrioid (~10%), clear cell (~10%), mucinous (~3%), and low-grade serous carcinoma (LGSC; <5%) [10]. The histopathologic classification of epithelial OC largely determines the clinical course of the disease. From this perspective, epithelial OC is divided into tumors with a good (type I) or poor (type II) prognosis. Type II OC is more prevalent as it comprises the most common HGSC and several less common OC subtypes, including carcinosarcoma and other mixed or undefined epithelial OC. Type II is characterized by frequent abnormalities in p53-related and homologous recombination (HR) DNA repair pathways, resulting in genome instability, and includes high-grade tumors with high proliferative potential and rapid progression, contributing to the late diagnosis at advanced stages [11,12]. In contrast, less common type I ovarian tumors include LGSC, clear cell carcinoma, endometrioid carcinoma, mucinous carcinoma, seromucinous carcinoma, and transitional cell carcinoma. Compared to the known genetic instability of type II tumors, type I tumors are relatively genetically stable. Type I OC are typically low-grade tumors with low proliferative activity and slow progression developing from benign lesions, particularly borderline tumors of the ovary, and, therefore, these tumors are more often diagnosed at an earlier stage [13,14].

Typically, OC develops in late adulthood, with a median age at diagnosis of 63 years [3]. Extremely early-onset ovarian tumors diagnosed between 18 and 30 years of age account for less than 5% of all OC cases (Figure 1) [3,15], and owing to their rarity, only a few studies have been conducted up-to-date. However, all the reports pointed out some striking differences between late- and early-onset adult OC, including genetic background, clinicopathologic features, or clinical outcomes compared with late-onset tumors (Table 1). Notably, while type II OC and particularly HGSC predominate in late-onset OC, approximately 50% of early-onset ovarian tumors are of germ-cell origin (representing a juvenile form of ovarian tumors diagnosed most frequently between the ages of 15 and 20 years), and only approximately 40% of the tumors belong to epithelial OC, particularly LGSC [16]. Correspondingly, late-onset OC is typically diagnosed at advanced stages, frequently with metastatic spread [3,17,18], whereas early-onset OC patients are typically diagnosed with localized disease. This would imply a better prognosis for early-onset OC patients [3,17,19], as also indicated by the significant survival advantage shown in a population-based study of OC patients [17]. However, the age at disease onset has not been confirmed as an independent stratifying factor concerning the prognosis for early-onset OC patients [19,20,21,22]. Particularly, in the long-term follow-up study, Gershenson et al. [22] observed a significantly worsened outcome, including both progression-free survival and overall survival in early-onset (≤35 years) compared with late-onset OC patients. Although LGSC generally have a more favorable prognosis in general [3,16,18], LGSC in early-onset OC patients have a worse prognosis and lower 5-year survival [22], providing another reason for separating this cohort from the majority of late-onset OC.

## 3. OC Risk Factors and Predisposition

The lifetime OC risk is about 1.1% in developed countries [3] and positively associated with lifetime ovulatory years (except for rare mucinous tumors) [27]. Moreover, the relative risk (RR) of OC development can increase approximately three times in cases of positive family epithelial OC history [28]. This is related to the high proportion of hereditary forms of OC, as >20% of OC patients are carriers of a GPV in an established cancer predisposition gene (Figure 1) [4,5,9]. Nevertheless, the frequency of GPV carriers differs substantially among patients with various histological subtypes. The highest proportion of GPVs are present in epithelial type II OC and, particularly in HGSC, the most prevalent late-onset OC, while the proportion of GPVs decreases in type I OC, including LGSC, and is the lowest in clear cell and mucinous type I carcinomas [29]. The predisposition to non-epithelial ovarian tumors is much less understood.

### 3.1. Established OC Predisposition Genes

The genetic predisposition to epithelial OC is well established, with major contributions from GPVs in genes coding for HR and mismatch repair (MMR) proteins (Table 2). The *BRCA1* and *BRCA2* HR genes are the most commonly germline-altered cancer predisposition genes responsible for the development of hereditary breast and ovarian cancer (HBOC) syndrome. GPVs in both genes could be found in nearly 20% of all OC patients [30]. The OC risk for *BRCA1* and *BRCA2* carriers rises significantly from the age of 35 and 45 years and reaches 58% and 29%, respectively [31,32]. An order of magnitude lower frequency but still significant risk is associated with carriers of GPVs in other HR genes, *RAD51D*, *RAD51C*, and *BRIP1*, having a life-time risk of epithelial OC of 20%, 15%, and 15%, respectively [33,34]. The median age of OC onset in *BRCA1* GPV carriers is significantly lower (53 years) compared to *BRCA2*, *RAD51D*, *RAD51C*, and *BRIP1* GPV carriers (59, 57, 62, and 65 years, respectively) or the general population; however, tumors before the age of 30 are rare, which is also reflected by the clinical management guidelines [31,33,34,35,36]. An increased risk of epithelial OC has also been documented for carriers of GPVs in MMR genes associated with Lynch syndrome (LS) (Table 2). The life-time risk varies in a range from a few percent (*MSH6* and *PMS2*) up to 20% (*MLH1*) and 38% (*MSH2*/*EPCAM*) [37]. However, while *MLH1*, *MSH2*, and *MSH6* are strongly associated with OC, the role of *PMS2* in OC predisposition is limited. Lynch syndrome OC patients are typically younger, with a median age at diagnosis of 43 and 46 years for *MSH2* and *MLH1*, respectively [38,39], and have also been described in early-onset OC patients [40]. In addition to the high-penetrance epithelial OC predisposition genes, GPVs in other genes involved in double strand break repair, including *ATM* and *PALB2*, have been associated with moderate risk to late-onset epithelial OC [41,42,43]. However, the clinical utility of the moderate-penetrance genes is low and largely depends on family OC history.

In addition, other rarely mutated genes are associated with non-epithelial ovarian tumors (Table 2). The GPVs in *STK11*, causing rare Peutz–Jeghers syndrome, are associated with a non-epithelial ovarian tumor risk exceeding 10% and the development of early-onset tumors in patients <30 years [44,45,46]. Only a few episodical reports document the development of early-onset epithelial OC in carriers of GPVs in *STK11* (Table 2) [4,41]. GPVs in *DICER1* have been linked to early-onset, non-epithelial sex cord-stromal ovarian neoplasms [47,48] accounting for nearly half of the stromal ovarian tumors [49]. However, *DICER1* appears to be exclusively characteristic of non-epithelial, early-onset ovarian tumors. Furthermore, GPVs in *SMARCA4* were associated with small cell carcinoma of the ovary hypercalcemic type (SCCOHT), a rare, aggressive OC similar to malignant rhabdoid tumors that primarily affects women under 40 years of age [50]. GPVs in *SMARCA4* have been reported recently in two early-onset OC patients; nevertheless, selectively in SCCOHT [51].

Generally, early cancer onset indicates the involvement of hereditary cancer predispositions [52,53]. Thus, one might anticipate an enrichment of GPV carriers in high-penetrance genes in early-onset OC patients. However, this assumption does not apply for early-onset epithelial OC patients, who were found to rarely carry GPVs, with the frequency not exceeding 10% (Figure 1) [4,7,8,9,38]. Nevertheless, only a limited number of studies focusing on the genetic predispositions of early-onset OC have been performed so far (Table 3).

**Table 2 ijms-24-17020-t002:** Established OC predisposition genes.

Gene	Heterozygote				Homozygote/ Compound Heterozygote [54]
	Associated OC Histotype	Absolute Risk for OC [35]	GPV Identified in Early-Onset OC	Other Associated Cancer Types [35]	
High penetrance					
BRCA1	Epithelial [41]	39–58%	Yes [4,9]	BC, PaC, PrC	FA-S
BRCA2	Epithelial [41]	13–29%	Yes [4]	BC, PaC, PrC, MM	FA-D1
BRIP1	Epithelial [41]	5–15%	Yes [4,9]	BC, CrC, EC	FA-J
DICER1	Sex cord-stromal [47]	NA	Yes [47,48]	DICER1 sy	-
MLH1	Epithelial [37,41]	4–20%	No	Lynch sy—CrC, EC, PaC	CMMRD
MSH2	Epithelial [37]	8–38%	Yes [40,52]	Lynch sy—CrC, EC, PaC	CMMRD
RAD51C	Epithelial [41]	10–15%	Yes [4]	BC	FA-O
RAD51D	Epithelial [41]	10–20%	No	BC	-
SMARCA4	SCCOHT [50]	NA	Yes [50,51]	Rhabdoid tumor predisposition sy	-
STK11	Non-epithelial [45]	>10%	Yes [46]	Peutz–Jeghers sy, BC, PaC, CrC	-
Moderate penetrance/Insufficient evidence					
ATM	Epithelial [41]	2–3%	Yes [4,9]	PaC	AT
MSH6	Epithelial [37]	1–13%	No	Lynch sy—CrC, EC, PaC	CMMRD
PMS2	Epithelial [37]	1–3%	Yes [40,52]	Lynch sy—CrC, EC	CMMRD
PALB2	Epithelial [42]	3–5%	No	BC, PaC	FA-N

AT, ataxia-telangiectasia; BC, breast cancer; CMMRD, constitutional mismatch repair deficiency syndrome; CrC, colorectal cancer; EC, endometrial cancer; FA, Fanconi anemia; GPV, germline pathogenic/likely pathogenic variant; MM, malignant melanoma; NA, not available; OC, ovarian cancer; PaC, pancreatic cancer; PrC, prostate cancer; SCCOHT, small cell carcinoma of the ovary hypercalcemic type; sy, syndrome.

This controversy was first acknowledged by Stratton et al. in 1999 [8], who tested only a few genetic loci (*BRCA1*, *MLH1*, *MSH2*, and a part of *BRCA2*) in the largest cohort of early-onset OC patients up to date. Since then, few studies have been conducted, but all of them confirmed a low frequency of GPVs among early-onset OC patients diagnosed <30 years [4,7,9,38,40,51,56,57]. Four recent larger studies, Carter et al. (2018) [9], Lhotova et al. (2020) [4], Flaum et al. (2023) [40], and Herold et al. (2023) [51], identified 2/147 (1.4%), 6/84 (7.1%), 4/77 (5.2%), and 3/83 (3.6%) GPV carriers in established high-penetrance OC predisposition genes (including 0.7%, 3.6%, 0%, and 0% mutations in *BRCA1*/*BRCA2*), respectively (Table 3). All four studies used the panel NGS approach, but with a different range of analyzed genes. While Carter [9] and Flaum [40] analyzed 15 and Herold [51] included 25 established cancer predisposition genes in their panel (Appendix A), Lhotova [4] used a much wider panel, targeting 219 established and candidate cancer predisposition genes. Despite the different design of the analysis, the similar results pointing to a very low frequency of GPV carriers in early-onset OC patients were strikingly similar. Interestingly, Flaum [40] associated GPVs in *MSH2* with early-onset OC, as 5.2% (4/77) of the patients carried the same *MSH2* GPV. This association was in coherence with the findings of another study focusing on OC in LS patients that included also three early-onset OC patients (Table 3) [52]; however, no other study further supported *MSH2*-association with early-onset OC. Due to the rarity of early-onset OC, further studies evaluating genetic predisposition to OC have included only a few OC cases with diagnoses at such a young age [52,55,58,59,60], leading to a limited understanding of the genetic factors underlying early-onset OC development. The overall lack of GPV carriers among early-onset OC patients is, however, evident and implies the need for the separation of this cohort from OC. Based on the frequency of GPVs in established OC predisposition genes, the cut-off age for distinguishing between early- and late-onset epithelial OC is around 30, as the frequency of *BRCA1* GPVs (a major genetic contributor to OC) starts to increase from age 35 [4].

### 3.2. Candidate OC Predisposition Genes

Candidate OC predisposition genes can be proposed based on their predisposition to other cancer types or associated diseases, which, however, have not yet been associated with OC (Table 4). The high-penetrance cancer predisposition genes *APC*, *BMPR1A*, *BAP1*, *FH*, *MEN1*, *PTEN*, *VHL*, *WT1*, and *TP53* (on top of their strong association with gastrointestinal tumors, melanoma, leiomyomatosis, multiple endocrine neoplasia, hamartomas, kidney tumors, breast cancer (BC), central nervous system tumors, and sarcomas, respectively) were also identified in OC patients and/or families [5,62,63,64,65,66,67,68,69,70], suggesting a potential wider cancer manifestation in these syndromes. Moreover, *BARD1* and *CHEK2* [5,69,70,71,72,73,74,75] were associated with a high to moderate risk of other cancer types, but their contribution to OC risk remains to be confirmed. Nevertheless, GPVs in both of these genes have been reported in several early-onset OC patients [4,9,55,58,59,60].

Furthermore, as the majority of established OC predisposition genes (Table 2) code for proteins involved in DNA repair and/or DNA damage response, the new candidate OC predisposition genes are often sought among genes involved in these pathways. GPVs in genes of the Fanconi anemia complex (including *FANCA*, *FANCC*, *FANCL*, *FANCM*, and *SLX4*), MRN complex genes (*MRE11*-*RAD50*-*NBN*), and other genes associated with DNA repair pathways have been reported in OC patients, suggesting their possible role in OC predisposition [5,41,51,55,68,75,76,77,78,79,80,81,82,83,84,85]. Moreover, new potential candidate genes emerged from complex sequencing studies of OC patients without being previously clearly established with any cancer, namely *ABRAXAS1* (also known as *FAM175A*), *CNKSR1*, and *PIK3C2G* [56,86,87]. However, it cannot be ruled out that all these more or less isolated findings may be coincidental and do not significantly affect the OC risk. Summarizing, the role of all the above-mentioned, candidate genes in early-onset OC is unclear, calling for further investigation in this field.

**Table 4 ijms-24-17020-t004:** Candidate OC predisposition genes.

Gene	GPVs Identified in Early-Onset OC	Associated Disease (Inheritance Mode) [54]	OC-Association Reported in
*ABRAXAS1 (FAM175A)*	No	-	[56,86]
*ATR*	No	Cutaneous telangiectasia and cancer sy (AD)	[78]
*APC*	Yes (early 30s) [63]	Familial adenomatous polyposis (AD)	[62,63]
*BAP1*	No	Melanoma (AD)	[68]
*BARD1*	Yes [6,25,55,59,60]	BC (AD)	[71]
*BLM*	No	Bloom sy (AR)	[5,76]
*BMPR1A*	Yes [64]	Juvenile polyposis sy, primary ovarian insufficiency (AD)	[64]
*BRAT*	Yes (in their 30s) [85]	Neurodevelopmental disorder (AR)	[85]
*CNKSR1*	No	-	[87]
*CDKN2A*	No	MM, MM-PaC sy (AD)	[5,72]
*CHEK2*	Yes [6,25,55,59,60]	BC (AD)	[5,73]
*ERCC3*	No	Trichothiodystrophy, xeroderma pigmentosum (AR)	[88,89]
*FANCA*	No	FA (AR)	[77,78]
*FANCC*	No	FA (AR)	[77]
*FANCL*	No	FA (AR)	[77]
*FANCM*	Yes [51]	-	[77,79]
*FH*	No	Leiomyomatosis and renal cell cancer (AD), fumarase deficiency (AR)	[68]
*MEN1*	No	Multiple endocrine neoplasia (AD)	[65]
*MRE11*	No	AT-like disorder (AR)	[81,84]
*NBN*	Yes [25]	Nijmegen breakage sy (AR)	[41]
*NF1*	No	Neurofribromatosis (AD)	[5]
*PIK3C2G*	No	-	[87]
*POLD1*	No	CrC, EC (AD)	[74]
*POLE*	No	CrC, EC (AD), IMAGE-I sy (AR)	[73]
*POLK*	No	-	[75]
*PTEN*	Yes [66]	Cowden sy (AD)	[66,67]
*RAD50*	No	Nijmegen breakage syndrome-like disorder (AR)	[84]
*RAD51B*	No	-	[5,82]
*RAD52*	No	-	[83]
*RAD54B*	No	-	[83]
*RAD54L*	No	-	[55]
*RB1*	No	Retinoblastoma (AD)	[5]
*RTEL1*	No	Dyskeratosis congenita (AD/AR), telomere-related pulmonary fibrosis, and/or bone marrow failure sy (AD)	[68]
*SLX4*	No	FA (AR)	[75,81]
*TP53*	Yes [51,69]	Li–Fraumeni sy (AD)	[5,69,70]
*TSC2*	No	Tuberous sclerosis (AD)	[68]
*VHL*	No	von Hippel–Lindau sy, pheochromocytoma (AD)	[68]
*WT1*	No	Wilms tumor (AD)	[68]
*XRCC3*	No	-	[82]

AD, autosomal dominant; AR, autosomal recessive; AT, ataxia-telangiectasia; BC, breast cancer; CrC, colorectal cancer; FA, Fanconi anemia; IMAGE, intrauterine growth restriction, metaphyseal dysplasia, adrenal hypoplasia congenita, and genitourinary abnormalities; MM, malignant melanoma; PaC, pancreatic cancer; sy, syndrome.

## 4. Alternative Approaches to Germline Genetic Testing in Early-Onset OC

The lack of identified GPVs in established and candidate cancer predisposition genes in early-onset OC patients could indicate a different disease-causing genetic basis compared to late-onset OC. Applying wider-scope germline whole exome sequencing (WES) or even whole genome sequencing (WGS) may provide new insight into early-onset OC genetics. WES has been used for the genetic analysis of OC (including some early-onset cases) [5,55,84]; however, the results have not shed light on an apparent, clinically relevant, new predisposition gene. Promisingly, WGS could identify yet missed variants (including deep intronic, untranslated regions, and copy number variants) in established and candidate or completely newly associated genes involved in OC predisposition [90,91,92]. Considering complex WES/WGS data analysis and interpretation, complementation with RNA-NGS could help to better understand the cancer-predisposition molecular mechanisms [93]. Nonetheless, these methods are currently not commonly used in clinical practice, as the limitations are not only the substantial cost of the testing and the massive abundance of the NGS data, but most importantly, the unclear clinical significance of the identified variants [94,95].

### 4.1. Alternative Ways of Cancer Predisposition Inheritance

Apart from applying new testing and analytical methods, various genetic and non-genetic factors (Figure 2) need to be considered in a complex analysis of early-onset OC predisposition. The association of proposed genetic components, as well as other factors predisposing to OC mentioned below has not been established for early-onset OC yet, but there are some pieces of evidence suggesting it could cause or contribute to early-onset cancer or to OC development in general.

In addition to autosomal dominant Mendelian inheritance, recessive inheritance may rarely contribute to the early-onset of OC, as exemplified by isolated cases of biallelic GPV carriers in *BRCA1* [96] or *PMS2* [52]. These sporadic cases suggest that recessive inheritance represents an uncommon cause of early-onset OC. Similarly, the development of de novo GPVs in *BRCA1* has been reported in rare cases of early-onset HBOC patients [97,98]. Also, a de novo GPV in *SMARCA4* has been observed in a patient with early-onset OC and other types of childhood/early-onset rhabdoid tumors [99]. The low frequency of de novo mutations among OC patients may have been biased by the HBOC genetic testing criteria, which used to prioritize patients with strongly positive family histories [100]. As early-onset OC patients lack a strong family cancer history [8], the de novo or compound heterozygous OC-predisposition mutations may be under-reported, and their identification, e.g., using a trio WES (analysis of the patient + her parents) similar to other rare childhood/early-onset diseases [101] may help to uncover the underlying genetic causes.

In addition, constitutional mosaicism of GPVs (with lower variant allele frequency in peripheral blood) in established/candidate genes with Mendelian inheritance may be a specific issue. Constitutional mosaicism of GPVs in *BRCA1* and *BRCA2* [98,102,103] has been previously reported in HBOC patients, suggesting that even the low-level mosaic GPVs in peripheral blood may be significant for the phenotype.

Finally, epigenetic inactivation can contribute to hereditary OC, as shown by promoter methylation of *BRCA1* [104], or *MLH1* in LS patients recently [105]. However, the prevalence of this phenomenon is largely unknown due to the limited data.

### 4.2. Family History and X-Linked Inheritance

Despite the low proportion of strong cancer family history in early-onset OC patients compared to patients with OC diagnosed > 30 years, the familiar form of early-onset OC still raises some important questions. Stratton et al. [8] described a slightly elevated OC risk, but also significantly enriched non-Hodgkin lymphoma and myeloma among first-degree relatives of early-onset OC patients (in the majority of cases without known germline genetic predisposition). Similarly, Rantala et al. [106] associated an increased incidence of early-onset (<40 years) OC and testicular cancer in patients whose aunts suffered from early-onset BC. On the other hand, Imbert-Bouteille et al. [107] found no relation between the occurrence of early-onset OC in families with *BRCA1*/*BRCA2* mutations and the age at diagnosis of BC or OC in their relatives. When focusing on the sex of the affected relatives, Stratton et al. [108] noticed that an affected woman’s sisters are at higher risk of disease than their mothers. In sync, Eng et al. described an X-linked association between prostate cancer in men and OC in their mothers and daughters [109]. These somewhat contradictory findings about the importance of family history in early-onset OC predisposition may point in the direction of an X-linked inheritance.

Apart from single nucleotide variants, X-linked OC predisposition can include complex genetic effects like X chromosome inactivation (XCI). Buller et al. [110] described a higher frequency of skewed XCI among OC patients, suggesting X-linked tumor suppressor genes or X-linked low penetrance susceptibility alleles affected by the inactivation pattern [111]. On the other hand, skewed XCI has been described at an increased frequency in BRCA1 mutation carriers compared with controls, and is associated with a statistically significant increase in age at diagnosis of breast and ovarian cancer in *BRCA1*/*BRCA2* GPV carriers [112]. In addition to the effect on the age of onset, molecular signatures of XCI were associated with clinical outcomes in epithelial OC, as patients with dysregulated XCI had shorter progression and overall survival than those with regulated XCI [113], suggesting a complex involvement of the X chromosome in OC.

### 4.3. Polygenic Inheritance

A complex polygenic inheritance stemming from an additive effect of multiple genetic variants may also be involved in early-onset OC predisposition (Figure 2). Polygenic inheritance could also explain the observed lack of positive family history in close relatives of early-onset OC patients [8]. A polygenic model of inherited predisposition to cancer was proposed by Qing et al. [114], who identified a higher burden of germline variants in protein-coding cancer hallmark genes predicted to alter the structure, expression, or function in early-onset patients compared to late-onset ones. They hypothesized that the early-onset carriers of more germline low-risk variants needed to harbor fewer somatic mutations for malignant transformation. Their hypothesis was supported by a significant association with several cancer types, including the OC.

Currently, hundreds of cancer-risk single nucleotide polymorphisms (SNPs) have been identified by genome-wide association studies, improving their understanding but not fully uncovering their polygenic heritability. These SNPs are considered causal or linked to causal variants. Using a polygenic risk score (PRS), some studies have recently shown a cumulative impact of SNPs in patients with several cancer diagnoses, including OC [115]. Especially serous OC showed association with PRS [116,117], and, interestingly, association was also found between PRS and early diagnosis of BC [118], together suggesting PRS might be associated with early-onset OC. In addition to PRS, another OC risk score evaluating epistatic gene interactions via chromosomal-scale length variation was proposed [119].

### 4.4. Di/Oligenic Inheritance

Since both monogenic and polygenic inheritance in OC have been demonstrated, the involvement of digenic to oligogenic inheritance might presumably also be a part of the OC genetic predisposition (Figure 2). In general, several di/oligogenic mechanisms have been suggested.

Firstly, a higher incidence and earlier onset of cancer have been proposed in carriers of two or more GPVs in high-penetrance cancer predisposition genes called multilocus inherited neoplasia allele syndrome (MINAS). However, *BRCA1*/*BRCA2*-MINAS patients did not develop OC significantly earlier than single GPV carriers [120], further disputing the *BRCA1*/*BRCA2* involvement in early-onset OC predisposition.

Secondly, a combination of a monogenic GPVs and a phenotype-modifying (e.g., age of onset) variant were reported. Carriers of GPVs in established OC-predisposition genes *BRCA1*/*BRCA2*, together with the newly associated gene *PPARGC1A*, were diagnosed at significantly earlier ages, suggesting *PPARGC1A* is a modifier of OC onset in *BRCA1*/*BRCA2* carriers [121]. Moreover, ethnically-specific modifiers were proposed to influence the phenotype in *BRCA1* GPV carriers [122], suggesting missing clues even in *BRCA1*/*BRCA2* otherwise well-established OC predisposition. On the other hand, focusing on *BRCA1*/*BRCA2* negative OC patients, Eng et al. [109] proposed one X-linked SNP in the *MAGEC3* gene to advance the age of OC onset by almost seven years. The OC risk or age at onset might also be modified by genetic variants in regulatory elements such as miRNA. Dysregulation of *BRCA1*/*BRCA2* functions in OC was reported in the absence of *BRCA1*/*BRCA2* GPVs stemming from miRNA dysregulation [123], and, especially, miR-146a polymorphism was associated with an earlier age of onset in *BRCA1*/*BRCA2*-negative HBOC patients [124].

Thirdly, a truly digenic disorder might be considered. Despite several clearly reported oligogenic disease associations, such as familial hemophagocytic lymphohistiocytosis, primary immunodeficiency, or familial hypercholesterolemia [125,126,127,128], the majority of proposed di/oligogenic allele combinations found in affected patients remain of uncertain significance [129]. Currently, we can only hypothesize about combinations of genetic variants that are separately nonpathogenic (e.g., missense) but together pathogenic when present in mutually interacting domains of proteins involved in OC predisposition, asking for a battery of functional tests and clinical investigation. Despite the currently unavailable clear association with early-onset OC, some evidence about the di/oligogenic inheritance involvement in cancer and, particularly, OC predisposition was proposed. Mouse model-based results suggest that oligogenic inheritance is also a part of cancer predisposition [130], supported by the first few reported clinical cases of digenic colorectal and gastric cancer in the OLIDA (OLIgogenic diseases DAtabase) [129].

### 4.5. Immune-Related Modifiers of OC

To further expand the complexity of OC predisposition, the immune system is also involved in cancer development (Figure 2). It has already been established that local chronic inflammation and autoimmune disease, including genetic immune dysregulations, predispose to certain types of cancer, such as autoimmune hepatitis-induced cirrhosis to hepatocellular carcinoma [131], Helicobacter pylori infection to gastric cancer [132], or inflammatory bowel disease to colorectal cancer [133]. Therefore, one could speculate that a systemic or ovarian-localized inflammatory condition may also, with time, lead to OC, possibly on an immunogenetic inherited basis. Currently, the knowledge of inherited immunity aspects associated with OC is limited, but a few examples have been described in the literature. Namely, systemic lupus erythematosus was reported to be associated with a higher incidence of cancer, including ovarian cancer [134]. Moreover, a significantly increased incidence of specific HLA-class II haplotypes (namely, DRB1*0301-DQA1*0501-DQB1*0201 and DRB1*1001-DQA1*0101-DQB1*0501) has been observed in OC patients, suggesting their role in OC pathogenesis [135].

### 4.6. Non-Genetic Factors

Similarly, genotype-environment interactions are a great unknown in the field of OC predisposition. The fact that the interaction of exposure to environmental carcinogens and constitutional genetics modulates cancer risk was first suggested in autosomal recessive Bloom and Werner syndromes caused by biallelic mutations in *BLM* and *WRN*, respectively [136]. Since then, several genotype-environment interactions have been recognized, including the combined presence of GPVs in *BAP1* and exposure to asbestos fibers, which together increase the risk of disease more than either component alone [137]. Moreover, the genotype-environment interaction may possibly modulate phenotypic heterogeneity, as shown in Birt–Hogg–Dubé syndrome caused by GPVs in *FLCN* or suggested for *DICER1* [138,139], which were also both reported in families with OC cases [47,48].

Moreover, lifestyle can also modify pre-existing OC predisposition. Namely, alcohol intake elevates the risk for OC [140], as well as exposure to smoking in childhood. Interestingly, smoking exposure was more likely associated with LGSC and non-serous OC [141], which are more frequent OC types among early-onset OC patients [16,17]. On the other hand, smoking has also been identified as a protective factor in endometroid and clear-cell OC [142]; however, smokers among early-onset OC patients have a short smoking history, limiting the effect of this risk factor.

Finally, we can also raise the question of whether survival differs in extremely young OC patients compared to histology-matched late-onset patients, as shown in early-onset LGSC OC patients with a less favorable prognosis and decreased 5-year survival [22]. Isolated studies point to some predictive markers associated with better survival in ovarian cancer patients, such as *PRDM1* variant rs2185379, which is suggested to be positively associated with the long-term recurrence-free survival of advanced OC [143]. On the other hand, the *EXO1* variant rs851797 was negatively associated with progression-free and overall survival in OC patients [144]. However, these isolated association studies typically deal with HGSC with late-onset and focus on patients from a specific population and should be interpreted with caution.

## 5. Conclusions

Despite being rare, early-onset OC represents a significant healthcare and socioeconomic problem, and its development is currently poorly understood. The risk of the disease can be modulated not only by genetics but also by environmental/lifestyle factors and/or their interactions. Supported by the rarity of the disease, it cannot be ruled out that early-onset OC (<30 years) is an unpredictable stochastic event caused just due to chance by random variables of different values.

Knowledge of the genetic factors could help identify women at risk of early-onset OC and might also be of predictive and prognostic value. Nevertheless, this is currently impossible, as we do not fully understand the factors associated with OC risk at such a young age. Since early-onset OC shows a significantly distinct germline mutation status compared to late-onset OC and the number of studies analyzing the GPVs in early-onset OC is limited, it is necessary to investigate early-onset OC patients in detail and separately from late-onset OC. Finding new approaches to germline genetic testing of early-onset OC patients is crucial for the identification of the expected yet unknown genetic causes, if present. These new insights could not only help to better understand the specifics of early-onset OC and provide new potential diagnostic and preventive targets for patient care and counseling, but they could also help to tailor the treatment modalities, thus improving the quality of life and survival of the early-onset OC patients.

## Figures and Tables

**Figure 1 ijms-24-17020-f001:**
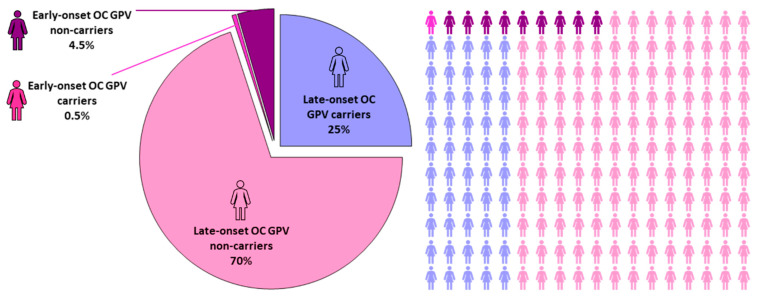
Frequency of germline pathogenic/likely pathogenic variants in established cancer predisposition genes in early-onset vs. late-onset OC patients.

**Figure 2 ijms-24-17020-f002:**
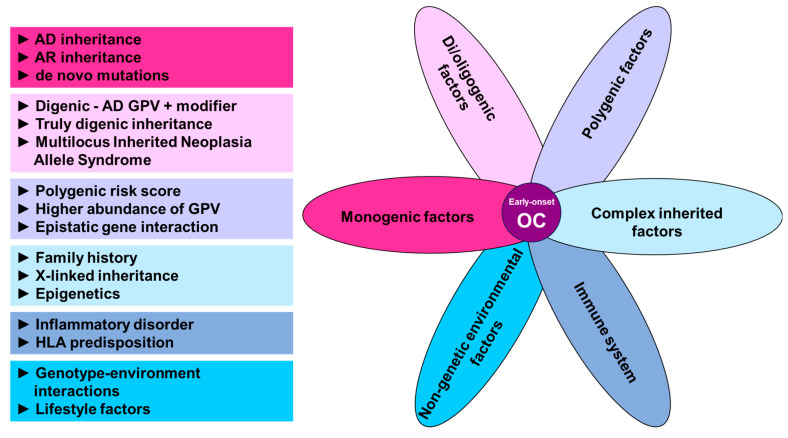
Proposed components of early-onset OC predisposition (AD, autosomal dominant; AR, autosomal recessive; GPV, germline pathogenic/likely pathogenic variant; OC, ovarian cancer).

**Table 1 ijms-24-17020-t001:** Characteristics of early-onset and late-onset OC.

Patients	Early-Onset OC	Late-Onset OC
Incidence in females	1.6/100,000 [3,15]	22.0/100,000 [3,15]
5-year relative survival rate	58–87% [17,19,20,23] -lower in LGSC [22]	Approx. 50% [3,17]
Clinicopathology		
Histology	~40% epithelial—LGSC prevails [16,17] ~50% germ-cell [16,24] ~10% sex cord-stromal [24,25]	~90% epithelial—HGSC prevails (70%) [13,26] ~6% sex cord-stromal [6] ~3% germ-cell [6]
Dominant tumor stage	Localized disease [3]	Distant disease [3]
Genetic predispositions		
GPV	Low <10% [7,8,9]	High >20% [4,5,9]

GPV, germline pathogenic/likely pathogenic variant; HGSC, high-grade serous carcinoma; LGSC, low-grade serous carcinoma; OC, ovarian cancer.

**Table 3 ijms-24-17020-t003:** Studies involving early-onset OC patients.

Study	Population	Study Details	No. of Tested Genes *	No. of All OC Patients	No. of Early-Onset OC Patients	Range of Early-Onset Patients’ Age at Dg.	Early-Onset OC Patients		
							No. of High-Penetrance GPV Carriers	GPVs in Established High-Penetrance OC Predisposition Genes	GPVs in Candidate OC Predisposition Genes
Stratton (1999) [8]	UK	Early-onset epithelial OC	4 **	169	169	13–30	0 ***	0	0
Carter (2018) [9]	US	OC	15	4439	147	6–30	2 (1.4%)	1×*BRCA1*; 1×*BRIP1*	3×*ATM*; 1×*BARD1*; 5×*CHEK2*
Lhotova (2020) [4]	CZ	OC	219	1333	84	15–30	6 (7.1%)	2×*BRCA1*; 1×*BRCA2*; 2×*RAD51C*; 1×*STK11*	1×*ATM*; 1×*BARD1*; 4×*CHEK2*; 1×*NBN*;
Herold (2023) [51]	GER	OC	25	206	83	13–30	3 (3.6%)	1× *BRIP1*; 2×*SMARCA4*	1×*FANCM*; 1×*MUTYH* het; 1×*PMS2*; 1×*TP53*
Flaum (2023) [40]	UK	Early-onset OC	15	77	77	15–30	4 (5.2%)	4×*MSH2*	1×*PMS2*
Bernards (2015) [38]	US	Early-onset OC	18	47	5	27–30	0	0	0
Felicio (2020) [55]	BRA	BRCA neg., TP53 neg. HBOC patients	WES	11	3	20–21	0	0	1×*CHEK2*
Da Costa (2020) [56]	BRA	HBOC	21	6	2	22–30	0	0	0
Boyd (2000) [57]	Jew	OC	2	189	1	25	0	0	0
Hajkova (2019) [58]	CZ	Synchronous EC and OC	73	22	1	29	0	0	1×*BARD1*
Jarhelle (2019) [59]	NOR	HBOC BRCA1/BRCA2 neg.	94	20	1	27	0	0	1×*CHEK2*
Risch (2001) [7]	US	OC	2	649	NA (96 <40 yo)	20-30	0	0	0
Koczkowska (2018) [60]	PL	OC	25	333	NA	NA	0	0	1×*CHEK2*
Pal (2005) [61]	US	epithelial OC	2	209	NA (11 <40 yo)	18-30	0	0	0
Ryan (2017) [52]	UK	LS assoc. preselected OC positive for LS-GPV	4	53	NA	24-30	2	2×*MSH2*	1×*PMS2* biallelic
Hajkova (2019) [58]	CZ	Synchronous EC and OC	73	22	1	29	0	0	1×*BARD1*

Dg, diagnosis; GPV, germline pathogenic variants; HBOC, hereditary breast and ovarian cancer; LS, Lynch syndrome; OC, ovarian cancer; NA, not available; No.; number; WES, whole exome sequencing; yo, years old; * The list of tested genes in each study is available in Appendix A; ** in the case of *BRCA2*, only the OC cluster region was analyzed; *** identified variants; *MLH1*: c.1853A > C/p.Lys618Thr and *MLH1*: c.1000G > C/p.Asp304His are not currently considered GPVs.

## Data Availability

The data presented in this study are available on request from the corresponding author.

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
