# Peer review of "Early-Onset Ovarian Cancer <30 Years: What Do We Know about Its Genetic Predisposition?"

_ijms, 2023, doi:10.3390/ijms242317020_

Round 1

Reviewer 1 Report

Comments and Suggestions for Authors

The manuscript submitted for review is an interesting discussion of genetic factors in young patients with ovarian cancer.

The manuscript meets the requirements for publication in IJMS.

My comments:

The authors write that there are epithelial OC and non-epithelial OC. In my opinion, if the tumor is nom-epithelial, it is not cancer. This is a malignant ovarian tumor. (e.g.: l.47,74)

There is a mess in the manuscript regarding citations. E.g. l.37 citations suddenly [24-26], l.125 [59,57,62]

This needs improvement

The manuscript is suitable for publication after minor corrections.

Reviewer 2 Report

Comments and Suggestions for Authors

This is a thorough and well written review on the subject of genetic predisposing factors and their role in early age onset of ovarian cancer with a focus on epithelial ovarian carcinomas.

The review covers an extensive literature providing some evidence for other genetic factors or modes of inheritance that predispose to early onset disease given the knowledge that the major hereditary breast and ovarian cancer predisposing genes (BRCA1 and BRCA2), less commonly associated genes (RAD51C, RAD51D and BRIP1, as examples) and other known genetic factors involved in colorectal cancer syndromes (MLH1, MSH2, etc) are less frequently involved in early stage disease (<30 years of age on onset). 

The study also highlights the paucity of research in this area of the role of heritable genetic factors in conferring risk to early onset ovarian cancer which are likely due to the rarity of disease presentation at a young age.

The review is timely and challenges the ovarian cancer research community to delve more deeply in the role of hereditary factors in early stage disease.

One observation that could be stressed in the review, is that the majority of high risk genetic factors, such as BRCA1, BRCA2, and others, have been identified in the contact of cancer syndromes that feature other cancer types where penetrance is higher for such types of cancer (ie beast, colorectal). 

Another observation is stressing the challenges in identifying other genetic predisposing factors given the rarity of the disease which might require large-scale international studies to achieve significance in identifying genetic markers of risk for heritable factors.
